

# Optimizing stretch-shortening cycle performance: effects of drop height and landing strategy on lower-limb biomechanics in drop jumps

Qin Zhang[1], Fei Li[1], Danielle Anne Trowell[2,3], Muzu Hou[1],
Zhenghe Qiu[2], Shiqin Chen[1] and Haifeng Ma[1]

[1] School of Athletic Performance, Shanghai University of Sport, Shanghai, China
[2] School of Exercise and Nutrition Sciences, Deakin University, Melbourne, Australia
[3] Centre for Sport Research, Institute for Physical Activity and Nutrition, Deakin University, Burwood, Victoria, Australia

## ABSTRACT

**Background:** The stretch-shortening cycle (SSC) enhances performance in jumping, sprinting, and changes of direction. Drop heights and landing strategies affect its efficiency. This study investigates the effects of varying drop heights and landing strategies (hip- *vs.* knee-dominant) on lower-limb stretch-shortening cycle performance during drop jumps (DJs), which involve a drop followed by an immediate vertical jump.

**Methods:** A three-dimensional (3D) motion capture system and force plate collected biomechanical data from 18 college athletes performing DJs with hip- and knee-dominant strategies at 30, 45, and 60 cm heights. A two-factor repeated measures analysis of variance (ANOVA) compared peak impact force, reactive strength index (RSI), leg stiffness ($K_{leg}$), joint stiffness ($K_{joint}$), joint angular displacement, change in joint moment, and joint work (positive, negative, net) across heights and strategies.

**Results:** Drop height significantly affected biomechanical variables ($p < 0.05$). Peak impact force and negative joint work increased from 30 cm to 60 cm, with the highest values at 60 cm. RSI, $K_{leg}$, $K_{joint}$, and net joint work peaked at 30 cm. Landing strategy significantly influenced outcomes ($p < 0.05$). The knee-dominant strategy had higher peak impact force, RSI, $K_{leg}$, knee angular displacement, change in knee moment, and ankle work, but lower net knee work, compared to the hip-dominant strategy, which showed higher hip angular displacement and hip work. A significant interaction was observed between drop height and landing strategy ($p < 0.05$). The knee-dominant strategy had greater RSI, $K_{leg}$, and positive ankle work at 30 cm, while the hip-dominant strategy had greater negative ankle work at 60 cm.

**Conclusion:** In DJs, SSC performance was optimised at a 30 cm drop height, with peak efficiency observed in the knee-dominant strategy. At 45 and 60 cm, SSC efficiency declined and knee energy dissipation increased, while the hip-dominant strategy may provide greater joint protection by increasing energy dissipation at the ankle. These findings suggest the knee-dominant strategy is best suited to 30 cm, whereas the hip-dominant strategy may enhance safety at higher drop heights.

Corresponding author
Haifeng Ma, mahf70@163.com

## INTRODUCTION

The stretch-shortening cycle (SSC) is a key mechanism that enhances performance in dynamic movements such as jumping, sprinting, and directional changes (*Nicol, Avela & Komi, 2006*). This process, characterized by a rapid transition of the muscle-tendon unit (MTU) from eccentric to concentric contractions, improves force output and explosive power, particularly in the lower limbs (*Laffaye & Wagner, 2013*; *Nicol, Avela & Komi, 2006*). The lower extremities, comprising multiple joints and associated MTUs, function as a complex "spring system" (*Kuitunen, Komi & Kyröläinen, 2002*), where muscle fascicles decouple from the MTU, resulting in lower fascicle velocities and greater tendon stretch. The tendon stretch stores elastic energy during the eccentric phase, which is then rapidly released during the concentric phase to augment force output (*Lichtwark & Wilson, 2006*; *Farris et al., 2016*; *Aeles et al., 2018*). The drop jump (DJ) is a widely used exercise for evaluating SSC efficiency, as it involves a rapid transition from impact absorption to explosive take-off. It reflects the coordinated interaction of muscle-tendon forces across the hip, knee, and ankle (*Peng, 2011*; *Zushi et al., 2022*). The DJ involves dropping from a platform, landing on the ground, and immediately performing a maximum vertical jump to optimize the SSC (*Bobbert, Huijing & van Ingen Schenau, 1987a*, *1987b*; *Peng, 2011*). As a key plyometric training tool, the DJ contributes to improved SSC utilization and overall athletic capabilities (*Di Giminiani & Petricola, 2016*; *Furuhashi et al., 2023*).

In the context of DJ performance, variables such as drop heights and landing strategies significantly influence biomechanical efficiency and SSC performance (*Horita et al., 2002*; *Moran & Wallace, 2007*). Drop heights govern the eccentric load at touchdown, with greater heights shown to increase neuromuscular pre-activation and lower limb stiffness (*Mrdaković et al., 2008*; *Hollville et al., 2019*), while facilitating muscle-tendon interactions that enhance elastic energy recoil and force potential (*Aeles et al., 2018*). In DJs, reactive strength index (RSI)–the ratio of jump height to ground contact time–quantifies this force production (*Flanagan & Comyns, 2008*). Research indicates that the optimal drop height for RSI generally falls between 30 to 60 cm (*Byrne et al., 2010*). However, excessively high drop heights may increase impact forces and injury risk without proportionate performance benefits (*Wang et al., 2021*). Furthermore, biomechanical factors such as ground reaction force, joint moment, work, and stiffness vary with drop heights, underscoring the complexity of determining an optimal drop height that balances performance enhancement with injury prevention (*Peng, 2011*). Thus, while RSI is a valuable metric, it may not fully capture biomechanical risks, emphasizing the need for a more comprehensive evaluation framework to identify a drop height that ensures both effective and safe training. Additionally, landing strategies influence these biomechanical variables, shaping SSC efficiency through body positioning, muscle activation patterns, joint motions, and force transmission (*Moran & Wallace, 2007*). Upon ground contact, the ankle initially absorbs impact forces, followed by rapid involvement of the knee and hip to distribute load and facilitate controlled deceleration (*Kotsifaki et al., 2021*). Adjusting the

range of motion (ROM) in the hip or knee joints during the landing process is believed to influence how impact forces are transmitted and dissipated throughout the lower limb, thereby affecting joint loading and the overall biomechanical response (*Moran & Wallace, 2007*; *Romanchuk, Del Bel & Benoit, 2020*).

*Seki et al. (2023)* were the first to compare the effects of hip-dominant and knee-dominant landing strategies. Hip-dominant landings involve greater hip flexion ROM and reduced knee flexion ROM, while knee-dominant landings exhibit the opposite pattern. The study demonstrated that each strategy distinctly alters mechanical work and energy distribution across the lower limbs. However, their study focused exclusively on the countermovement jump (CMJ) and did not examine whether similar strategies influence DJ performance. Additionally, *Di Giminiani et al. (2020)* reported that as drop height increased up to a maximum of 60 cm, athletes tended to adopt a knee-dominant strategy. This suggests an interaction between drop heights and landing strategies in DJ performance. However, adopting a knee-dominant strategy at higher drop heights may increase the risk of lower limb injuries (*Peng, Kernozek & Song, 2011*). The lack of distinction between hip-dominant and knee-dominant strategies in previous DJ studies contributes to uncertainty surrounding the role of these strategies, indicating the need for further investigation to clarify how they influence both performance and injury risk in dynamic drop tasks. Therefore, the aim of this study was to investigate the effects of drop heights and hip- *vs.* knee-dominant strategies on lower limb SSC performance during the DJ and to explore their interaction. This analysis was conducted to enhance understanding of SSC mechanisms in the DJ, provide a theoretical basis for designing targeted training programs, and offer practical insights into injury prevention.

## METHODS

### Participants

Eighteen male collegiate athletes (age: 23.22 ± 2.10 years; height: 174.47 ± 7.35 cm; body mass: 76.03 ± 11.66 kg) were recruited, including seven soccer players, five handball players, and six badminton players. These sports were selected because they involve jumping, sprinting, and change of direction movements, which rely on effective utilization of the SSC (*Nicol, Avela & Komi, 2006*). Male athletes were specifically chosen to avoid the significant biomechanical differences in drop and jumping characteristics between males and females that could influence SSC performance (*Baus, Harry & Yang, 2020*). Participants had not experienced any lower-limb musculoskeletal injury in the 6 months prior to the experiment. After being fully informed of the testing procedures and potential risks, each participant signed an informed consent form. This study was approved by the Ethics Committee of Shanghai University of Sport (approval number: 102772024RT126).

### Study design

Upon arrival at the laboratory, participants first completed a preparation phase, including a standardized 10-min warm-up protocol (*e.g.*, low-intensity jogging and dynamic stretching exercises). The researcher demonstrated the experimental movements in detail, provided clear instructions, and conducted several practice trials to ensure participants

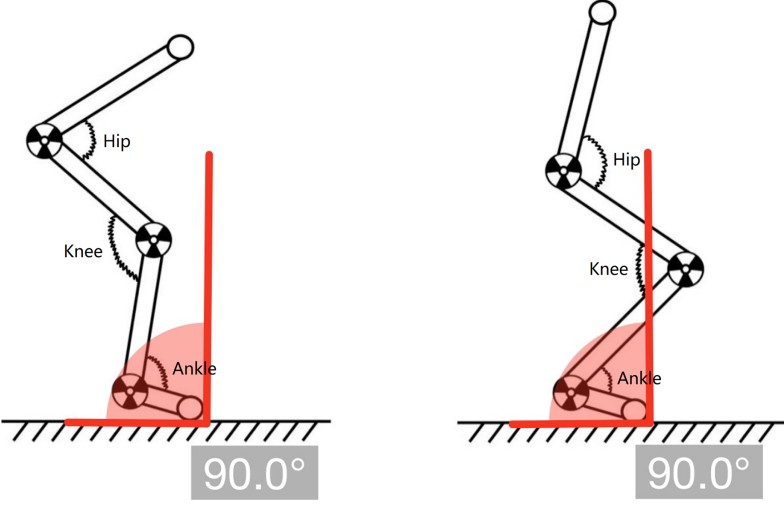

**Figure 1 Hip-dominant and knee-dominant landing strategies.** The left panel shows a hip-dominant strategy, and the right panel shows a knee-dominant strategy.

fully understood the procedures. Participants were instructed to position themselves on the drop box with their arms crossed over their chest to minimize upper limb involvement (*Ward et al., 2019*). Participants initiated each trial by stepping off the box with their dominant leg, ensuring no active push-off, to maintain an initial vertical velocity close to zero. Participants were instructed to adopt one of two randomized landing strategies upon ground contact: the hip-dominant or knee-dominant strategy. Immediately after landing, participants performed a vertical jump with maximal effort, ensuring full extension of the hip and knee at takeoff without pausing. The hip-dominant strategy was characterized by posterior hip displacement while maintaining the knees behind the toes, whereas the knee-dominant strategy involved anterior knee displacement beyond the toe position (*Seki et al., 2023*). A two-dimensional motion analysis system (Dartfish™, Dartfish Inc., Fribourg, Switzerland) identified strategies (*Ortiz et al., 2016*), as shown in Fig. 1. During the testing phase, participants performed DJ tests from three different heights (30, 45, and 60 cm) in randomized order, using either the hip-dominant or knee-dominant landing strategy.

## Data acquisition

Thirty-six reflective markers and T-frames were placed on participants' pelvis and lower extremities to define the pelvis, thigh, shank, and foot segments. Kinematic data were collected using a 10-camera three-dimensional motion capture system (Vicon Metrics Ltd., Oxford, UK; 200 Hz), while ground reaction forces were recorded *via* two force platforms (9290AA; Kistler, Winterthur, Switzerland; 1,000 Hz). Joint kinetic data (*e.g.*, joint moments, joint work) captured throughout the ground contact phase enabled a comprehensive analysis of energy dissipation and regeneration during the SSC. During DJ tests, the criteria for a valid test included the following: ensuring that the body did not shift

significantly upward or forward during the drop phase, adhering to the prescribed hip-dominant or knee-dominant strategy, and keeping the arms crossed over the chest throughout the entire process. Five valid trials were completed for each height-strategy combination, with a 1-min rest between trials and a 3-min rest between conditions (*Kümmel et al., 2018*; *Rajic et al., 2020*). The average of five trials was used for subsequent analyses.

## Data processing

Kinematic data in the sagittal plane were filtered using a fourth-order Butterworth low-pass filter with a cut-off frequency of 7 Hz in Visual3D software (*Sun et al., 2019*). Initial foot contact and toe-off events were determined when the vertical ground reaction force exceeded or fell below 10 N, respectively. The movement phases were divided based on the body's center of mass (COM), which was estimated using a six-segment human body model (right and left thigh, lower-leg, and foot segments) (*Peng, 2011*). The COM location was calculated by weighting the positions of body segments according to their mass, using marker data from anatomical landmarks and established anthropometric models (*Winter, 2009*). The contact phase was defined as the period from foot contact to toe-off and was further divided, at the joint level, into an eccentric phase (from foot contact to the lowest position of the COM) and a concentric phase (from the lowest position of the COM to toe-off) (*Peng, 2011*). The flight phase was defined as the period from toe-off to subsequent foot contact.

The peak impact force was defined as the maximum vertical ground reaction force during the eccentric phase (*Peng, 2011*). The RSI was calculated as the ratio of jump height ($H_f$) to contact time ($T_c$), as shown in the following equation:

$$RSI = \frac{H_f}{T_c}$$

$K_{leg}$ was determined by the ratio of vertical ground reaction force at the lowest point of the COM (GRFi) to the vertical displacement of the COM during the eccentric phase ($\Delta y$), as shown in the following equation:

$$K_{leg} = \frac{GRFi}{\Delta y}$$

Joint stiffness ($K_{joint}$) was calculated as the ratio of joint torsional deformation moment ($\Delta M$) to angular displacement ($\Delta \theta$) during the eccentric phase, as shown in the following equation:

$$K_{joint} = \frac{\Delta M}{\Delta \theta}$$

Net joint work ($W_{net}$) was the sum of positive joint work ($W_{pos}$) and negative joint work ($W_{neg}$), reflecting the utilization efficiency of elastic energy at the joint level (*Sun et al., 2019*), as shown in the following equation:

$$W_{net} = W_{pos} + W_{neg} .$$

## Statistical analysis

The analysis was conducted using SPSS 25.0 software (IBM, Armonk, NY, USA). All data were expressed as mean ± standard deviation. A two-factor (drop height × landing strategies) repeated measures analysis of variance (ANOVA) was performed to examine the effects of different drop heights (30, 45, and 60 cm) and landing strategies (hip-dominant and knee-dominant) on biomechanical outcome variables. Mauchly's test of sphericity was used to assess whether the data met the assumption of sphericity. When the sphericity assumption was violated (*i.e.*, Mauchly's test was significant, $p < 0.05$), the Greenhouse-Geisser correction was applied to adjust the degrees of freedom. When significant main effects and interaction effects were found, pairwise comparisons with Bonferroni correction were conducted for *post-hoc* analysis. The level of statistical significance was set at $p < 0.05$. Effect sizes were reported using partial eta squared ($\eta^2$), with values of 0.01–0.09, 0.09–0.25, and >0.25 representing small, medium, and large effects, respectively (*Richardson, 2011*).

# RESULTS

Table 1 summarizes biomechanical variables (mean ± SD and F values) across drop heights (30, 45, and 60 cm) and landing strategies (hip-dominant and knee-dominant) for main and interaction effects.

## The main effect of drop heights

Drop heights significantly influenced multiple biomechanical variables. Peak impact force (F = 63.90, $p < 0.001$, $\eta^2 = 0.79$) increased progressively with height, with 30 cm significantly lower than 45 and 60 cm ($p < 0.001$), and 45 cm lower than 60 cm ($p < 0.001$). RSI (F = 9.137, $p = 0.001$, $\eta^2 = 0.350$) and $K_{leg}$ (F = 20.082, $p < 0.001$, $\eta^2 = 0.542$) were higher at 30 cm than at 45 cm ($p \leq 0.016$) and 60 cm ($p \leq 0.010$), with $K_{leg}$ also higher at 45 cm than 60 cm ($p = 0.012$). Hip stiffness (F = 4.849, $p = 0.014$, $\eta^2 = 0.222$) and knee stiffness (F = 10.196, $p = 0.002$, $\eta^2 = 0.375$) were greater at 30 cm than 45 cm ($p \leq 0.038$) and/or 60 cm ($p \leq 0.008$), while ankle stiffness (F = 9.248, $p = 0.001$, $\eta^2 = 0.352$) was higher at 30 cm than 45 cm ($p = 0.051$) and 60 cm ($p = 0.005$). Hip (F = 10.057, $p < 0.001$, $\eta^2 = 0.372$), knee (F = 7.346, $p = 0.002$, $\eta^2 = 0.302$), and ankle angular displacements (F = 15.099, $p = 0.001$, $\eta^2 = 0.470$) were lower at 30 cm compared to 60 cm ($p \leq 0.011$), with marginal differences between 45 and 60 cm ($p = 0.060$–0.073). Net hip work (F = 31.765, $p < 0.001$, $\eta^2 = 0.651$), net knee work (F = 65.320, $p < 0.001$, $\eta^2 = 0.793$), and net ankle work (F = 66.611, $p < 0.001$, $\eta^2 = 0.797$) were highest at 30 cm, decreasing with height ($p \leq 0.002$). Negative hip (F = 10.702, $p < 0.001$, $\eta^2 = 0.386$), knee (F = 59.492, $p < 0.001$, $\eta^2 = 0.778$), and ankle work (F = 40.534, $p < 0.001$, $\eta^2 = 0.705$) increased with height ($p \leq 0.003$). No significant effects were observed for changes in hip or ankle moment, and while change in knee moment showed a main effect (F = 3.928, $p = 0.029$, $\eta^2 = 0.188$), *post-hoc* tests revealed no specific differences.

**Table 1 Test metrics under different drop heights and landing strategies.**

| | Landing strategies | Drop heights | | | Main effect (F-value & p-value) | | Interaction (F-value & p-value) |
|---|---|---|---|---|---|---|---|
| | | 30 cm | 45 cm | 60 cm | Drop height | Landing strategies | |
| Peak impact force (BW) | HD | 1.64 ± 0.32 | 2.11 ± 0.56(a) | 2.65 ± 0.50(ab) | **F = 63.90** | **F = 6.51** | F = 1.25 |
| | KD | 1.84 ± 0.40 | 2.29 ± 0.50(a) | 2.68 ± 0.61(ab) | $\boldsymbol{p < 0.001^{**}}$ | $\boldsymbol{p = 0.021^*}$ | $p = 0.298$ |
| Reactive strength index (m.s$^{-1}$) | HD | 0.69 ± 0.16 | 0.65 ± 0.14 | 0.65 ± 0.15 | **F = 9.14** | **F = 62.31** | **F = 7.11** |
| | KD | 0.84 ± 0.20 | 0.76 ± 0.17(a) | 0.73 ± 0.16(a) | $\boldsymbol{p = 0.001^*}$ | $\boldsymbol{p < 0.001^{**}}$ | $\boldsymbol{p = 0.003^*}$ |
| Leg stiffness (BW·m$^{-1}$) | HD | 5.04 ± 1.42 | 4.60 ± 0.94(a) | 4.27 ± 0.74(ab) | **F = 20.08** | **F = 7.45** | **F = 4.12** |
| | KD | 5.70 ± 1.61 | 4.86 ± 1.22(a) | 4.51 ± 1.02(a) | $\boldsymbol{p < 0.001^{**}}$ | $\boldsymbol{p = 0.014^*}$ | $\boldsymbol{p = 0.025^*}$ |
| Hip stiffness (Nm·kg$^{-1}$·deg$^{-1}$) | HD | 0.058 ± 0.026 | 0.053 ± 0.017 | 0.055 ± 0.017 | **F = 4.85** | F = 3.25 | **F = 7.01** |
| | KD | 0.075 ± 0.042 | 0.058 ± 0.026(a) | 0.060 ± 0.033(a) | $\boldsymbol{p = 0.014^*}$ | $p = 0.089$ | $\boldsymbol{p = 0.009^*}$ |
| Hip angular displacement (deg) | HD | 49.40 ± 9.50 | 51.13 ± 8.72 | 51.98 ± 7.00 | **F = 10.06** | **F = 21.53** | **F = 7.90** |
| | KD | 39.70 ± 12.03 | 44.50 ± 9.56(a) | 48.19 ± 9.68(ab) | $\boldsymbol{p < 0.001^{**}}$ | $\boldsymbol{p < 0.001^{**}}$ | $\boldsymbol{p = 0.002^*}$ |
| Change in hip moment (Nm·kg$^{-1}$) | HD | 2.70 ± 0.94 | 2.61 ± 0.55 | 2.77 ± 0.64 | F = 1.27 | F = 2.71 | F = 0.06 |
| | KD | 2.54 ± 0.73 | 2.40 ± 0.70 | 2.61 ± 0.70 | $p = 0.286$ | $p = 0.118$ | $p = 0.942$ |
| Knee stiffness (Nm·kg$^{-1}$·deg$^{-1}$) | HD | 0.018 ± 0.007 | 0.017 ± 0.007 | 0.014 ± 0.007(ab) | **F = 10.20** | **F = 93.13** | F = 1.50 |
| | KD | 0.034 ± 0.012 | 0.031 ± 0.009 | 0.027 ± 0.007(ab) | $\boldsymbol{p = 0.002^*}$ | $\boldsymbol{p < 0.001^{**}}$ | $p = 0.242$ |
| Knee angular displacement (deg) | HD | 64.31 ± 11.02 | 64.96 ± 13.97 | 65.54 ± 8.98 | **F = 7.35** | **F = 43.60** | **F = 7.67** |
| | KD | 70.40 ± 16.69 | 75.13 ± 13.40(a) | 79.63 ± 13.10(ab) | $\boldsymbol{p = 0.002^*}$ | $\boldsymbol{p < 0.001^{**}}$ | $\boldsymbol{p = 0.002^*}$ |
| Change in knee moment (Nm·kg$^{-1}$) | HD | 1.16 ± 0.38 | 1.11 ± 0.50 | 0.95 ± 0.53 | **F = 3.93** | **F = 113.88** | F = 0.30 |
| | KD | 2.30 ± 0.60 | 2.30 ± 0.63 | 2.16 ± 0.68 | $\boldsymbol{p = 0.029^*}$ | $\boldsymbol{p < 0.001^{**}}$ | $p = 0.740$ |
| Ankle stiffness (Nm·kg$^{-1}$·deg$^{-1}$) | HD | 0.021 ± 0.011 | 0.017 ± 0.006(a) | 0.014 ± 0.004(a) | **F = 9.25** | **F = 10.86** | F = 1.04 |
| | KD | 0.032 ± 0.020 | 0.025 ± 0.013(a) | 0.024 ± 0.015(a) | $\boldsymbol{p = 0.001^*}$ | $\boldsymbol{p = 0.004^*}$ | $p = 0.365$ |
| Ankle angular displacement (deg) | HD | 35.73 ± 12.48 | 41.97 ± 8.30(a) | 42.72 ± 6.91(a) | **F = 15.10** | F = 0.73 | F = 0.87 |
| | KD | 37.66 ± 12.20 | 41.85 ± 11.23(a) | 45.35 ± 10.10(a) | $\boldsymbol{p = 0.001^*}$ | $p = 0.406$ | $p = 0.428$ |
| Change in ankle moment (Nm·kg$^{-1}$) | HD | 0.65 ± 0.23 | 0.70 ± 0.16 | 0.61 ± 0.15 | F = 0.23 | **F = 39.79** | F = 1.98 |
| | KD | 1.00 ± 0.25 | 0.95 ± 0.27 | 0.99 ± 0.31 | $p = 0.796$ | $\boldsymbol{p < 0.001^{**}}$ | $p = 0.154$ |
| Net hip work (J·kg$^{-1}$) | HD | 0.34 ± 0.27 | 0.20 ± 0.25(a) | 0.09 ± 0.23(ab) | **F = 31.77** | F = 2.33 | F = 0.67 |
| | KD | 0.27 ± 0.21 | 0.16 ± 0.22(a) | 0.06 ± 0.22(ab) | $\boldsymbol{p < 0.001^{**}}$ | $p = 0.145$ | $p = 0.520$ |
| Negative hip work (J·kg$^{-1}$) | HD | −0.83 ± 0.22 | −0.82 ± 0.22 | −0.89 ± 0.23 | **F = 10.70** | **F = 36.69** | **F = 4.37** |
| | KD | −0.56 ± 0.23 | −0.68 ± 0.27(a) | −0.74 ± 0.25(a) | $\boldsymbol{p < 0.001^{**}}$ | $\boldsymbol{p < 0.001^{**}}$ | $\boldsymbol{p = 0.021^*}$ |
| Positive hip work (J·kg$^{-1}$) | HD | 1.18 ± 0.29 | 1.03 ± 0.28(a) | 0.98 ± 0.28(a) | **F = 8.52** | **F = 27.24** | **F = 7.54** |
| | KD | 0.83 ± 0.27 | 0.84 ± 0.27 | 0.81 ± 0.25 | $\boldsymbol{p = 0.001^*}$ | $\boldsymbol{p < 0.001^{**}}$ | $\boldsymbol{p = 0.002^*}$ |
| Net knee work (J·kg$^{-1}$) | HD | 0.18 ± 0.30 | 0.01 ± 0.32(a) | −0.22 ± 0.32(ab) | **F = 65.32** | **F = 23.66** | F = 1.67 |
| | KD | 0.05 ± 0.33 | −0.19 ± 0.29(a) | −0.46 ± 0.31(ab) | $\boldsymbol{p < 0.001^{**}}$ | $\boldsymbol{p < 0.001^{**}}$ | $p = 0.213$ |
| Negative knee work (J·kg$^{-1}$) | HD | −1.09 ± 0.35 | −1.26 ± 0.49(a) | −1.55 ± 0.40(ab) | **F = 59.49** | **F = 108.48** | **F = 3.31** |
| | KD | −1.67 ± 0.30 | −1.99 ± 0.47(a) | −2.28 ± 0.53(ab) | $\boldsymbol{p < 0.001^{**}}$ | $\boldsymbol{p < 0.001^{**}}$ | $\boldsymbol{p = 0.048^*}$ |
| Positive knee work (J·kg$^{-1}$) | HD | 1.27 ± 0.24 | 1.26 ± 0.31 | 1.33 ± 0.30(a) | **F = 4.67** | **F = 78.86** | F = 1.82 |
| | KD | 1.72 ± 0.32 | 1.79 ± 0.37 | 1.81 ± 0.39(a) | $\boldsymbol{p = 0.016^*}$ | $\boldsymbol{p < 0.001^{**}}$ | $p = 0.178$ |
| Net ankle work (J·kg$^{-1}$) | HD | 0.35 ± 0.15 | 0.21 ± 0.15(a) | 0.14 ± 0.14(ab) | **F = 66.61** | **F = 25.20** | F = 0.53 |
| | KD | 0.51 ± 0.16 | 0.39 ± 0.14(a) | 0.30 ± 0.13(ab) | $\boldsymbol{p < 0.001^{**}}$ | $\boldsymbol{p < 0.001^{**}}$ | $p = 0.594$ |

(Continued)

| Table 1 (continued) | | | | | | | |
|---|---|---|---|---|---|---|---|
| | Landing strategies | Drop heights | | | Main effect (F-value & p-value) | | Interaction (F-value & p-value) |
| | | 30 cm | 45 cm | 60 cm | Drop height | Landing strategies | |
| Negative ankle work (J·kg⁻¹) | HD | −0.35 ± 0.20 | −0.48 ± 0.21(a) | −0.54 ± 0.18(ab) | **F = 40.53** | F = 2.71 | **F = 3.44** |
| | KD | −0.34 ± 0.16 | −0.40 ± 0.17(a) | −0.46 ± 0.16(ab) | **p < 0.001**\*\* | p = 0.118 | **p = 0.044**\* |
| Positive ankle work (J·kg⁻¹) | HD | 0.69 ± 0.18 | 0.69 ± 0.17 | 0.68 ± 0.16 | **F = 5.77** | **F = 42.20** | **F = 8.07** |
| | KD | 0.85 ± 0.17 | 0.79 ± 0.16(a) | 0.76 ± 0.17(a) | **p = 0.017**\* | **p < 0.001**\*\* | **p = 0.001**\* |

**Note:**

Significant level (\*$p < 0.05$, \*\*$p < 0.001$). Significant main and interaction effects are indicated by bold letters and numbers. a indicates comparison with the 30 cm drop height; b indicates comparison with the 45 cm drop height; BW, body weight; HD, hip-dominant strategy; KD, knee-dominant strategy.

### The main effect of landing strategies

Landing strategies significantly influenced multiple biomechanical variables. Peak impact force ($F = 6.51$, $p = 0.021$, $\eta^2 = 0.277$) was higher with the knee-dominant strategy than the hip-dominant strategy ($p < 0.001$). RSI ($F = 62.314$, $p < 0.001$, $\eta^2 = 0.786$) and $K_{leg}$ ($F = 7.450$, $p = 0.014$, $\eta^2 = 0.305$) were greater with the knee-dominant strategy than the hip-dominant strategy ($p \leq 0.014$). Hip stiffness ($F = 3.247$, $p = 0.089$, $\eta^2 = 0.160$) was marginally higher with the knee-dominant strategy ($p = 0.089$), while hip angular displacement ($F = 21.529$, $p < 0.001$, $\eta^2 = 0.559$) was lower with the knee-dominant strategy ($p < 0.001$). No significant effects were found for change in hip moment. Knee stiffness ($F = 93.132$, $p < 0.001$, $\eta^2 = 0.846$), knee angular displacement ($F = 43.595$, $p < 0.001$, $\eta^2 = 0.719$), and change in knee moment ($F = 113.877$, $p < 0.001$, $\eta^2 = 0.870$) were higher with the knee-dominant strategy ($p < 0.001$). Ankle stiffness ($F = 10.858$, $p = 0.004$, $\eta^2 = 0.390$) and change in ankle moment ($F = 39.788$, $p < 0.001$, $\eta^2 = 0.701$) were greater with the knee-dominant strategy ($p \leq 0.004$), while no significant effects were found for ankle angular displacement. Negative hip work ($F = 36.687$, $p < 0.001$, $\eta^2 = 0.683$) and positive hip work ($F = 27.239$, $p < 0.001$, $\eta^2 = 0.616$) were higher with the hip-dominant strategy ($p < 0.001$), while no significant effects were found for net hip work. Negative knee work ($F = 108.478$, $p < 0.001$, $\eta^2 = 0.865$) and positive knee work ($F = 78.859$, $p < 0.001$, $\eta^2 = 0.823$) were higher with the knee-dominant strategy ($p < 0.001$), while net knee work ($F = 23.657$, $p < 0.001$, $\eta^2 = 0.582$) was higher with the hip-dominant strategy ($p < 0.001$). Net ankle work ($F = 25.196$, $p < 0.001$, $\eta^2 = 0.597$) and positive ankle work ($F = 42.200$, $p < 0.001$, $\eta^2 = 0.713$) were higher with the knee-dominant strategy ($p < 0.001$), while no significant effects were found for negative ankle work.

### The interaction between drop heights and landing strategies

Significant interaction effects between drop heights and landing strategies were observed for key biomechanical variables related to SSC performance. Figure 2 displays bar plots illustrating these interaction effects. RSI demonstrated a significant interaction effect ($F = 7.112$, $p = 0.003$, $\eta^2 = 0.295$). Within the knee-dominant strategy, RSI at 30 cm was significantly higher than at 45 cm ($p = 0.015$) and 60 cm ($p = 0.005$), while no significant differences were observed across heights in the hip-dominant strategy. Additionally, the
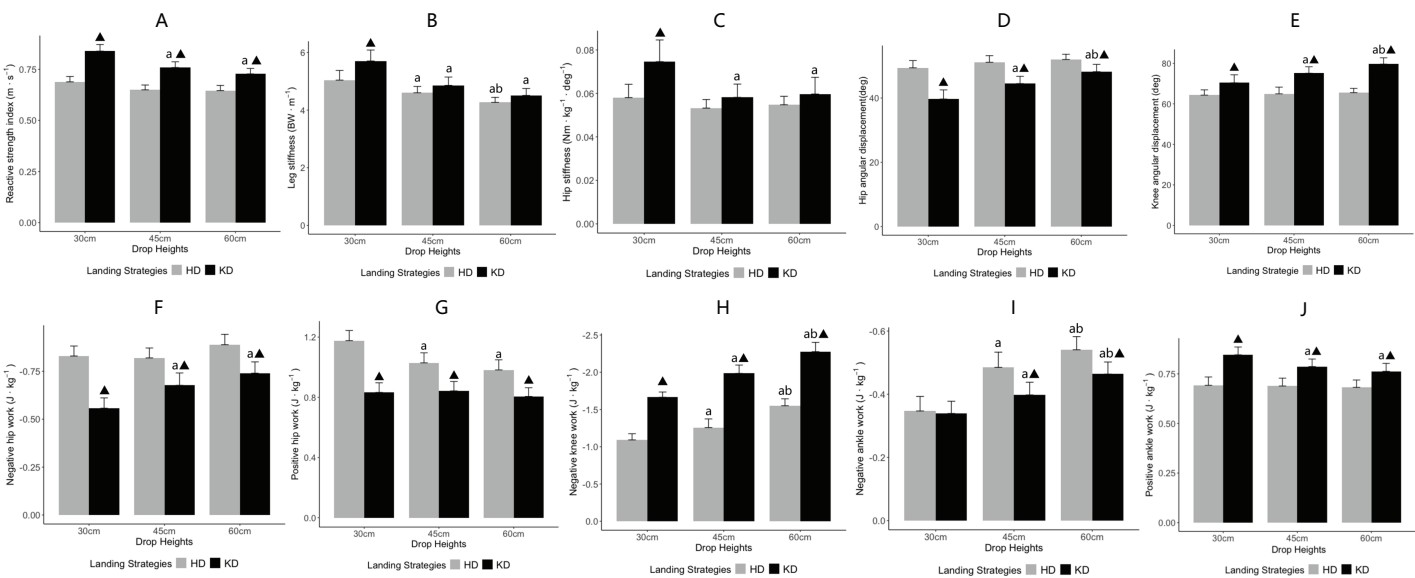

**Figure 2 The interaction between drop heights and landing strategies.** Test metrics (A–J) for the significant interaction between drop heights and landing strategies. 'a' indicates comparison with the 30 cm drop height ($p < 0.05$); 'b' indicates comparison with the 45 cm drop height ($p < 0.05$); '▲' indicates comparison with the hip-dominant strategy ($p < 0.05$). HD, hip-dominant strategy; KD, knee-dominant strategy.

knee-dominant strategy produced higher RSI values than the hip-dominant strategy at all heights ($p < 0.001$). $K_{leg}$ showed a significant interaction effect (F = 4.119, $p = 0.025$, $\eta^2 = 0.195$). In the knee-dominant strategy, stiffness at 30 cm was significantly higher than at both 45 and 60 cm ($p = 0.001$). In the hip-dominant strategy, 30 cm was also significantly higher than 45 cm ($p = 0.043$) and 60 cm ($p = 0.007$). Additionally, at 30 cm, the knee-dominant strategy produced greater $K_{leg}$ than the hip-dominant strategy ($p = 0.004$). Hip stiffness demonstrated a significant interaction effect (F = 7.006, $p = 0.009$, $\eta^2 = 0.292$). Within the knee-dominant strategy, stiffness at 30 cm was significantly higher than at 45 cm ($p = 0.013$) and 60 cm ($p = 0.010$), while no significant differences were observed across heights in the hip-dominant strategy. At 30 cm, hip stiffness was greater in the knee-dominant than the hip-dominant strategy ($p = 0.024$). Hip angular displacement also showed a significant interaction (F = 7.904, $p = 0.002$, $\eta^2 = 0.317$). In the knee-dominant strategy, 30 cm resulted in significantly lower displacement than both 45 cm ($p = 0.025$) and 60 cm ($p = 0.001$), whereas no significant differences were observed across heights in the hip-dominant strategy. Across all drop heights, the hip-dominant strategy produced greater angular displacement than the knee-dominant strategy ($p < 0.05$). Positive ankle work showed a significant interaction effect (F = 8.074, $p = 0.001$, $\eta^2 = 0.322$). In the knee-dominant strategy, values at 30 cm were significantly higher than at 45 cm ($p = 0.012$) and 60 cm ($p = 0.014$), while no significant differences were observed across heights in the hip-dominant strategy. Across all heights, the knee-dominant strategy produced greater positive ankle work than the hip-dominant strategy ($p < 0.05$).

Negative hip work (F = 4.367, $p = 0.021$, $\eta^2 = 0.204$) and positive hip work (F = 7.536, $p = 0.002$, $\eta^2 = 0.307$) both showed significant interaction effects. Within the

knee-dominant strategy, negative hip work at 30 cm was significantly lower than at 45 cm ($p = 0.014$) and 60 cm ($p = 0.001$). In the hip-dominant strategy, positive hip work at 30 cm was significantly higher than at 45 cm ($p = 0.004$) and 60 cm ($p = 0.003$). Across all heights, the hip-dominant strategy produced greater negative and positive hip work than the knee-dominant strategy ($p < 0.05$). Knee angular displacement showed a significant interaction effect (F = 7.674, $p = 0.002$, $\eta^2 = 0.311$). Within the knee-dominant strategy, displacement at 60 cm was significantly greater than at 30 and 45 cm (both $p = 0.002$), while no significant differences were observed across heights in the hip-dominant strategy. At all drop heights, the knee-dominant strategy produced greater knee angular displacement than the hip-dominant strategy ($p < 0.05$). Negative knee work also demonstrated a significant interaction effect (F = 3.313, $p = 0.048$, $\eta^2 = 0.163$). In the knee-dominant strategy, negative work at 60 cm was significantly greater than at 30 and 45 cm (both $p < 0.001$). Similarly, in the hip-dominant strategy, 60 cm produced significantly greater negative work than 30 and 45 cm (both $p < 0.001$). Across all heights, the knee-dominant strategy resulted in higher negative knee work than the hip-dominant strategy ($p < 0.001$). Negative ankle work showed a significant interaction effect (F = 3.438, $p = 0.044$, $\eta^2 = 0.168$). In the hip-dominant strategy, values at 60 cm were significantly higher than at 30 and 45 cm (both $p < 0.001$). In the knee-dominant strategy, 60 cm was also higher than 30 cm ($p = 0.003$) and marginally higher than 45 cm ($p = 0.052$). At 60 cm, negative ankle work was significantly greater in the hip-dominant strategy compared to the knee-dominant strategy ($p = 0.033$).

## DISCUSSION

This study investigated the effects of drop heights and landing strategies on lower-limb SSC performance during DJs. The results showed significant impacts of each factor on various biomechanical responses, including impact loading characteristics, lower limb reactive capacity (such as RSI), and mechanical characteristics (such as $K_{leg}$, joint-specific stiffness, angular displacement, change in moment, and energy management strategies involving negative-positive work transitions). The interaction between drop heights and landing strategies also affected reactive capacity (RSI) and mechanical characteristics (such as $K_{leg}$, hip stiffness, hip and knee angular displacement, positive and negative work of the hip and ankle, and negative knee work). The discussion addresses these findings from three perspectives: drop heights, landing strategies, and their interaction effects.

### Effects of drop heights on stretch-shortening cycle performance

The study found 30 cm to be the optimal drop height for male collegiate athletes performing a DJ as it maximised biomechanical parameters (except for peak impact force and changes in hip and ankle moments). This contrasts with the findings of *Mrdaković et al. (2008)*, who reported that the optimal drop height for DJ performed by long-trained national-level male football players was 60 cm, where RSI reached its peak. Although the present study did not assess muscle-tendon behavior, *van Ingen Schenau, Bobbert & de Haan (1997)* suggests that an increased drop height raises the amount of negative work and elastic energy stored in tendinous tissue during landing. This should, in turn, facilitate the
release of more elastic energy during takeoff. However, if the magnitude of muscle force (represented by the contractile element in the conceptual muscle model) is limited, the work is dissipated as heat rather than being stored in the tendinous tissue (*van Ingen Schenau, Bobbert & de Haan, 1997*; *Farris & Sawicki, 2012*; *Hollville et al., 2019*). This diminishes MTU elastic energy storage and stretch reflex efficiency, ultimately reducing SSC performance (*van Ingen Schenau, Bobbert & de Haan, 1997*). Elevated impact forces at greater jump heights also increase the risk of lower limb injuries, including anterior cruciate ligament tears and ankle sprains (*Beynnon et al., 2005*; *Walsh et al., 2004*). *Wang et al. (2021)* also found that male college athletes exhibited the highest $K_{leg}$ and the lowest peak impact force at a 30 cm drop height (within the 30–50 cm range). Thus, the male college athletes' lower strength levels likely explain their lower optimal drop height compared to elite athletes.

In addition, we found that joint stiffness and net joint work were significantly higher during 30 cm DJs compared to 45 and 60 cm DJs. Net hip and ankle work remained positive but gradually decreased with increasing drop height. In contrast, net knee work became negative as the drop height increased to 45 and 60 cm, indicating a shift toward predominant energy dissipation at the knee joint. Notably, while changes in hip and ankle moments were unaffected by drop heights, their stiffness decreased with drop height due to increased angular displacement ($K_{joint} = \Delta M/\Delta\theta$). This contrasts with the knee, where increased angular displacement and a reduced change in knee moment–reflected by a significant main effect of drop heights indicating a decreasing trend despite no differences between heights (Table 1)–resulted in reduced stiffness. Although joint kinematics differ from kinematics at the muscle-tendon level, anatomical features and behavior of muscles and tendons may explain differences in the mechanical characteristics of lower limb joints (*Aeles et al., 2018*; *Aeles & Vanwanseele, 2019*; *Hollville et al., 2019*). As drop height increases, the hip angular displacement significantly increases, which further activates the hamstrings and gluteus maximus, enhancing lower limb neuromuscular control (*Zushi et al., 2022*). Meanwhile, the ankle plantarflexor (*e.g.*, gastrocnemius medialis) exhibit quasi-isometric behavior, minimizing lengthening and thereby maintaining consistent force output. This is enabled by the Achilles tendon, which absorbs the increased mechanical load through greater stretching, protecting the muscle fibers from excessive strain and reducing force dissipation (*Kubo et al., 2007*; *Hollville et al., 2019*). In contrast, the knee's shorter patellar tendon, with its lower stiffness and higher sensitivity to viscosity, slightly limits its ability to absorb impact energy compared to the Achilles tendon, thereby forcing the knee extensor (*e.g.*, vastus lateralis) also to dissipate energy through eccentric activation during landing, which further increases negative work (*Hollville et al., 2019*). Under greater loading (*e.g.*, higher drop height), the knee extensor (*e.g.*, vastus lateralis) exhibits greater amplitude and velocity of lengthening (*Hollville et al., 2019*). This increased muscle lengthening velocity reduces force-generating capacity, likely contributing to the observed reduction in knee moment change. Thus, for the participants in this study, the 30 cm drop height balances maximal hip and ankle energy output with minimal impact loading and knee energy dissipation, optimizing DJ performance while reducing injury risk.

**Effects of landing strategies on stretch-shortening cycle performance**

In addition to drop heights, landing strategies significantly affect SSC performance during the DJ. Our results indicate that landing strategies significantly affect hip and knee angular displacement. The hip-dominant strategy exhibited greater hip angular displacement, while the knee-dominant strategy showed greater knee angular displacement; ankle angular displacement remained unaffected. This confirms the distinct nature of these strategies. The knee-dominant strategy resulted in higher $K_{leg}$ and RSI, while the hip-dominant strategy reduced peak impact force, highlighting a trade-off between performance and safety. The hip-dominant strategy activates the hamstrings and gluteus maximus to a greater extent than the knee-dominant strategy, enhancing neuromuscular control of the lower limbs, as supported by previous research (*Zushi et al., 2022*). Increased hip angular displacement provides mechanical advantages for hamstrings to counteract quadriceps-induced anterior tibial translation (*Blackburn & Padua, 2008*). Limited tibial anterior shift converts the knee's shear impact (anteroposterior) into a more vertical load, reducing peak stress on the knee (*Vignos et al., 2020*). Concurrently, hamstring co-contraction improves knee stability, distributing impact energy through muscle-tendon units rather than passive joint structures (*Chen et al., 2022*). The hip-dominant strategy, with a slight trunk lean and posterior hip shift, may bring the impact force closer to the body' COM, thereby shortening the moment arm and lowering the peak impact force (*Markström et al., 2020*).

In contrast, the knee-dominant strategy exhibited a higher peak impact force, but was also associated with greater RSI and $K_{leg}$, increased stiffness at the hip, knee, and ankle, and greater net ankle work. *Wang et al. (2021)* showed that higher $K_{leg}$ and ankle stiffness enhance SSC performance. Our findings highlight the ankle's crucial role in SSC optimization, as indicated by the greater net ankle work observed. Additionally, we found that the knee-dominant strategy leads to a greater change in ankle moment, while ankle angular displacement remained similar across strategies. This suggests that increased ankle stiffness primarily results from an icnreased ankle moment. Larger ankle plantarflexor forces bring them closer to isometric contraction, enhancing tendon lengthening and shortening, optimizing elastic energy storage and utilization (*Farris & Sawicki, 2012*; *Aeles et al., 2018*; *Hollville et al., 2019*). *Daley, Felix & Biewener (2007)* noted that the knee-dominant strategy improves the ankle's ability to store and release energy. Thus, the superior SSC performance of the knee-dominant strategy may be attributed to the efficiency of the ankle MTU. The knee-dominant strategy also showed higher hip stiffness and smaller hip angular displacement, with no significant changes in hip moment. These findings suggest that greater hip stiffness is associated with reduced angular displacement, while net hip work remains unchanged-indicating poorer SSC performance at the hip, likely due to shorter, stiffer tendons with limited energy storage capacity (*Wade, Lichtwark & Farris, 2018*; *Romanchuk, Del Bel & Benoit, 2020*). Regarding the knee joint, the knee-dominant strategy produced higher stiffness than the hip-dominant strategy, driven by increases in both knee angular displacement and change in knee moment. The substantial rise in moment change–approximately 2.1 times from hip-dominant to knee-dominant strategies–far exceeds the 1.16-fold increase in angular displacement,

explaining the stiffness increase (Table 1). However, compared to the hip-dominant strategy, the knee-dominant strategy results in more negative knee work and less net knee work. Given the anatomical features and behaviour of the knee joint's MTU, an excessive knee moment may increase joint pressure, which does not necessarily improve SSC performance and could increase the risk of knee injuries (*Moran & Wallace, 2007*; *Leppänen et al., 2017*). Therefore, the knee-dominant strategy may rely more on the ankle than the hip or knee for SSC performance during DJs. Future studies should further investigate MTU behaviour to clarify these joint-specific contributions.

## Interactive effects of drop heights and landing strategies on stretch-shortening cycle performance

This study explored the interactive effects of drop heights and landing strategies on SSC performance during DJs, revealing how these factors collectively influence various biomechanical parameters such as reactive capacity and mechanical characteristics. The results indicate that at a drop height of 30 cm, the knee-dominant strategy outperforms the hip-dominant strategy in terms of RSI, $K_{leg}$, hip joint stiffness, and positive ankle work. However, due to the lack of significant difference in net hip work between the hip-dominant and knee-dominant strategies, the influence of the hip joint on SSC performance is relatively minor. Instead, the greater positive ankle work observed at the 30 cm height likely plays a more significant role in optimizing SSC performance. In contrast, the hip-dominant strategy at a 60 cm drop height shows significantly higher hip angular displacement, positive and negative hip work, and negative ankle work compared to the knee-dominant strategy, but knee negative work is lower than in the knee-dominant strategy. Specifically, in the hip-dominant strategy, athletes shifted some of the impact force from the ankle to the hip by increasing hip joint angle displacement, this reducing knee joint load (*Daley, Felix & Biewener, 2007*). However, our results show that as drop height increases to 45 and 60 cm, the peak impact force significantly increased, but hip negative work remained unchanged. Although knee negative work increases, it remains significantly lower than in the knee-dominant strategy across all drop heights. This suggests that, in the hip-dominant strategy, the energy absorption capacity of the hip and knee may be inadequate to fully cope with the increased impact at higher drop heights, leading the ankle joint to take on additional negative work. The high compliance of the Achilles tendon is suggested as a potential factor that allows it to absorb mechanical energy during drop (*Hollville et al., 2019*). However, due to lower ankle stiffness in the hip-dominant strategy, this absorption primarily results in energy dissipation (negative work) (*Wang et al., 2021*). Thus, at higher drop heights, the hip-dominant strategy likely dissipates most impact forces through the ankle joint to protect the lower limbs from injury.

This study is not without its limitations. First, we included only male athletes to control for potential sex-based differences, which limits the generalizability of our findings. Due to common strength differences between sexes, females may have a lower optimal drop height compared to males. Second, different training backgrounds may influence optimal DJ conditions, with highly trained athletes potentially benefiting from higher drop heights

(*e.g.*, above 30 cm), while less-trained individuals may require lower drop heights. Third, although this study elucidates SSC performance from a biomechanical perspective, it did not investigate muscle-tendon behavior, such as muscle fascicle changes or tendon elastic energy dynamics, thereby restricting deeper mechanistic insights. Future research should explore how SSC performance varies across sexes and training levels, incorporating muscle-tendon analyses to enhance understanding.

## CONCLUSIONS

A DJ is an exercise involving a drop landing followed immediately by a maximal vertical jump. This study identified 30 cm as the optimal drop height to balance elastic performance and safety during the DJ exercise. Drop heights exceeding 30 cm were associated with reduced elastic energy efficiency and increased energy dissipation, particularly at the knee joint. In addition, a hip-dominant strategy was found to protect lower extremity joints by reducing peak impact force, while a knee-dominant strategy enhanced SSC performance but increased the risk of knee injuries. The interaction between drop heights and landing strategies revealed that the knee-dominant strategy exhibited optimal SSC at 30 cm; however, its performance benefits diminished at higher drop heights. At 60 cm, the hip-dominant strategy might dissipate most of the impact forces through the ankle joint, helping to protect the lower limbs from injury. Based on these findings, athletes are advised to perform DJs from a height of 30 cm or less using a knee-dominant strategy to optimise SSC performance, while remaining mindful of the associated higher injury risk. At higher drop heights, the hip-dominant strategy may be a safer choice to reduce injury risk during DJs.

## ACKNOWLEDGEMENTS

The authors thank all the athletes who participated in this study. The authors also acknowledge the testing personnel for their dedicated support throughout the study.

### Funding

This work was supported by the 2023 Local College Capacity Building Program of the Shanghai Science and Technology Commission (No. 23010504300) and the Key Laboratory of Human Performance at Shanghai University of Sport (No. 11DZ2261100). The funders had no role in study design, data collection and analysis, decision to publish, or preparation of the manuscript.

### Grant Disclosures

The following grant information was disclosed by the authors:
Shanghai Science and Technology Commission: 23010504300.
Key Laboratory of Human Performance at Shanghai University of Sport: 11DZ2261100.

### Competing Interests

The authors declare that they have no competing interests.

## Author Contributions

- Qin Zhang conceived and designed the experiments, performed the experiments, analyzed the data, prepared figures and/or tables, authored or reviewed drafts of the article, and approved the final draft.
- Fei Li conceived and designed the experiments, authored or reviewed drafts of the article, and approved the final draft.
- Danielle Anne Trowell analyzed the data, authored or reviewed drafts of the article, and approved the final draft.
- Muzu Hou performed the experiments, analyzed the data, prepared figures and/or tables, and approved the final draft.
- Zhenghe Qiu analyzed the data, authored or reviewed drafts of the article, and approved the final draft.
- Shiqin Chen performed the experiments, prepared figures and/or tables, and approved the final draft.
- Haifeng Ma conceived and designed the experiments, authored or reviewed drafts of the article, and approved the final draft.

## Human Ethics

The following information was supplied relating to ethical approvals (*i.e.*, approving body and any reference numbers):

This study was approved by the Ethics Committee of Shanghai University of Sport (approval number: 102772024RT126).

## Data Availability

The raw data is available in File S1. The raw data includes physiological data such as age, height, body mass, and body weight of the participants, as well as SSC performance metrics such as peak impact force, reactive strength index, leg stiffness, joint stiffness, joint angular displacement, change in joint moment, and joint work (negative, positive, and net). The data is categorized by drop heights (30, 45, 60 cm) and landing strategies (hip-dominant and knee-dominant). This data was used for a two-way repeated measures ANOVA to analyze the effects of drop heights and landing strategies on various SSC performance metrics.

## Supplemental Information

Supplemental information for this article can be found online at http://dx.doi.org/10.7717/peerj.19490#supplemental-information.

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
