# Peer review of "Optimizing stretch-shortening cycle performance: effects of drop height and landing strategy on lower-limb biomechanics in drop jumps"

_PeerJ, doi:10.7717/peerj.19490_

## Round 0.1 · original submission · Major Revisions

Based on the reviewer comments, I invite you to revise and resubmit the manuscript. The reviewers found the work interesting, and it seems possible for you to address all concerns.

·

Basic reporting

I have no concerns with the items for this section.

Experimental design

The study rationale is laid out well, the design is appropriate and well explained. Please see specific comments in attachment for minor suggestions.

Validity of the findings

The results are meaningful and well presented in the text. Please see specific comments in the attachment for minor suggestions.

·

Basic reporting

The manuscript is generally written very well and is appealing in structure and format. The introduction in particular reads very well and contains the necessary information. However, I have some major concerns about the wording used in the manuscript, in particular the interchanging of drop jump landings and stretch-shortening cycle movements, the interpretation of joint data for inferences about muscle-tendon dyanmics/elastic energy storage, and the data that is missing such as joint angles, ankle joint data, etc. I address these in detail below with plenty of important references that support my comments. With sufficient rephrasing and adding missing results, I believe this manuscript will make a nice contribution to the literature and be of use to the readership.


Major concerns
* * *
1) At the muscle-tendon level, there is a very big difference between a drop jump in which the goal is to land safely with no follow-up movement, and a counter-movement drop jump after which the goal is to perform a jump (or other movement) following the landing (Aeles & Vanwanseele, 2019). The latter is what is typically mostly associated with stretch-shortening cycles. It is unclear in the current introduction which movement type the authors are mostly focussing on as drop jumps and stretch-shortening cycles are used together here. I would advise making this clearer as the mechanisms behind these movements are very different also from an injury perspective (Holville et al., 2019, Konow et al., 2012, Werkhausen et al., 2017; Dick et al., 2021 Proc B). From the introduction it seems that the focus is on drop jumps and not so much on counter-movements, however in the methods on L 115-116 the authors highlight that athletes were chosen from sports that clearly use a post-landing movement involving different stretch-shortening cycle muscle-tendon dynamics (including differences in muscle activation).

2) Given that in any landing, the ankle joint is the first to encounter forces and therefore needs to deal with substantial joint torques and is the first to provide energy dissipation, I am surprised not to read anything about the ankle joint. Why do you only focus on knee and hip strategies? At the very least, I would recommend the authors to at least discuss the ankle joint and the differences in joint torques/energy dissipation between the ankle, knee, and hip joint and to address (perhaps based on torques etc) why the knee and hip should be of more importance in these landing strategies.

Other concerns
* * *
L71: “Neuromuscular activation”, is there any evidence (references) that landing height influences pre-activation, i.e., does a greater height lead to earlier activation? It seems to me this is what you suggest here. If so, please provide a reference, if not, then I would suggest rewriting this.
L72: There are several articles showing that muscle fascicles are decoupled from the muscle-tendon unit (e.g. Lichtwark & Wilson, 2006 JEB), resulting in lower muscle fascicle velocities leading to increase force potential (Aeles et al., 2017 JAP) and greater tendon stretch for elastic energy recoil (Farris et al., 2016), which is the predominant mechanism behind greater force potential.
L73-77: You use a reference from 2017 to define RSI, suggesting it was only established then. However, on line 77 you cite research from 2010 that found the optimal height for maximising RSI, suggesting RSI was already established earlier. I would recommend citing on line 74 the earliest work defining RSI.
L98: “landing height increased within a 60 cm range”, this is not very clear to me, especially the “within” and “range” are confusing. Do you mean, above 60 cm?
L204: The wording here is a bit confusing as it seems that the parameter that is statistically compared is the landing height. I would recommend changing the wording saying that it is the peak impact force that is compared (e.g.: “comparisons of landing heights showed that the peak impact force at 60 cm was …”) and adopt this throughout the results section.
L268: Here and elsewhere, change the text to “male collegiate athletes”.
L269-270: Did this study also only look at drop jump landing with no follow-up movement? As the differences would be big as I highlighted previously. The statement “at which the SSC is most effective” makes me wonder. What is the definition of an “effective SSC”? Is this not to maximise the follow-up movement’s joint torques/COM acceleration/… or is it only about energy dissipation here? This should be made clearer.
L272-273: This depends, as not all muscles undergo a stretch-shortening cycle (Aeles & Vanwanseele, 2019). Also, at some point the muscles cannot produce enough force to fully stay isometric and there may not be further or even less tendon stretching, reducing the elastic energy storage (Holville et al., 2018; Farris & Sawicki, 2012).
L292: Here again the authors mix “jumping” with “drop landing”, which as I mentioned several times are very different and cannot be so simply compared.
L300: In knee-dominant landing these muscles will also be activated. Do you mean that they are more active in hip vs knee strategy? Please make this clear in the text.
L301-303: Another reason to include the joint angles in this manuscript.
L303-304: Can you elaborate on this? It is typically the tendon, much more than the muscle fascicles, that absorb the impact force (see references in my other comments). Long fascicles do not contribute, in my opinion, much to energy absorption. Typically, fascicles shorten or restrict lengthening velocities during dissipation tasks (Konow, Holville, etc references).
L306-307: Net joint work does not say much about elastic energy, unless I am missing something. There could be zero absorption and very high net work, which does not mean elastic energy storage. Since you did not present both negative and positive net work separately, which I think would be very valuable, this cannot be determined.
L309-315: I very strongly disagree with these statements as joint angles say almost nothing about muscle or muscle fascicle lengths. They are completely decoupled. I have provided plenty of references throughout this review that show this.
L316-319: This should be removed as you did not investigate what is happening at the muscle-tendon level and therefore cannot justify this statement.
L331-334: I do not follow this. If greater landing heights generate greater impact forces, then should the differences between the strategies not increase with greater landing height? How are these related and why would it diminish? This should be made clearer in the text.
L337-339: But you asked your participants specifically to use both strategies, so how does this statement hold relative to your study design and data?
L343-345: Even better would be to assess the SSC at the muscle and tendon level as now inferences are made from joint data to the muscle-tendon behaviour, which is wrong.
L356-357: Also here it should be changed to emphasise the difference between a drop landing with no follow-up movements and a stretch-shortening cycle exercise (drop jump with immediate counter-movement jump).

Experimental design

Major concerns
* * *
1) Please explain why only males were included in the study. This significantly lowers the impact of your study given that it now excludes 50% of the population, and it cannot be generalised to female athletes. The authors should include in the methods on L113 why only males were included.

2) I am surprised to see that the authors decide to investigate knee and hip strategies (omitting the ankle) but then focus on ankle and knee joint analyses (omitting the hip). Why do you not also report the results from the hip joint given that your whole aim is to compare hip and knee strategies? There is likely also a large effect on the hip joint between the two strategies? Additionally, the ankle may go through different joint motions and moments between the two strategies, which may also explain part of the results. The authors should also report and analyse the ankle joint angle and joint moments to ensure they were not different between the two strategies.

3) The task and instructions should be described more clearly. In particular, what the athletes were told to do after the landing as this now does not seem controlled. I.e., bounce back up immediately to a fully extended position, or stay in the low position? Were joint work etc measured over the entire period? This instruction, i.e. what happened immediately after dissipating the energy will have a large effect on the joint work, potentially affecting the reported values.


Other concerns
* * *
Figure 2: I am not sure if this figure adds much to the manuscript. However, I leave this to the authors/editor to decide.
L159-160: I would recommend the authors make it explicit that they are talking about “joint eccentric” and “joint concentric” phases as the muscle has very different timings for these phases. I would change this throughout the manuscript. Additionally, I believe the authors should raise this point in the discussion, i.e. that at the muscle and tendon level, very different things are happening compared to the joint level (see my earlier references).
L159-166: COM is mentioned here often, but it has not been defined yet. How did you determine the COM?
L191: are there no earlier and perhaps more appropriate references for these values?

Validity of the findings

Major concerns
* * *
1) I am missing the kinematic data on the two different mechanisms. I assumed after reading the introduction that the two strategies would be based on the joint angles at the knee and hip. I was surprised to see this was not the case even though the authors collected all the marker data and analysed joint angles for determining joint stiffness and work done. Is using the knee beyond or behind the toes not a far inferior method compared to using the joint angles? Additionally, why are the joint angles not also presented in the results? I would highly recommend doing this as it will allow the reader to assess how well your athletes were able to differentiate the two different strategies. This is important for correctly interpreting the other results.

2) I am VERY surprised to see that the knee joint stiffness is greater in the knee-dominant strategy, where I would expect more knee joint ROM, leading to a lower stiffness. However, given that the authors did not report joint angles, nor joint moments, this cannot be determined. I highly recommend to also report the joint angles and moments, together with joint stiffness and net work for the hip joint.


Other concerns
* * *
Table 1: I would consider to show in the table for landing height what the post-hoc effect was, i.e. which of the 3 landing heights were significantly different from each other. Additionally, can knee joint stiffness not be measured/reported with greater resolution? Given the magnitude, only 2 significant numbers seems low.
Figure 3 is unreadable due to low quality and small size, which could be an issue with how the journal has compiled the PDF.
L287-290, this did not seem to happen in the study by Holville et al. (2018) from a landing of 25 to 50 cm in the gastrocnemius muscles in “only” recreationally active men. The ankle joint already dissipates part of the energy before it reaches the knee joint. If the, relatively weaker (!) gastrocnemii are strong enough to stay isometric, would we not expect the knee extensor muscles to be able to do this as well? Especially in collegiate male athletes?

---

## Round 0.2 · Minor Revisions

Both reviewers were happy with the revision, but there were a few remaining comments. Please revise the manuscript to address those comments.

**Language Note:** The review process has identified that the English language must be improved. PeerJ can provide language editing services - please contact us at [email protected] for pricing (be sure to provide your manuscript number and title). Alternatively, you should make your own arrangements to improve the language quality and provide details in your response letter. – PeerJ Staff

·

Basic reporting

My main remaining concern is that the manuscript needs a "once over" by a native English speaker to make sure diction, tense, etc. are correct. Anything done in the past should be past-tense ("aimed" vs "aims" for example in the Introduction - there are many other examples throughout the paper).

The title needs to be more descriptive as to the general categorization of dependent variables investigated. Also, add the word "the" after "during" (or, conversely, add a "s" to jump to make it "jumps").

Experimental design

I have no remaining issues with the experimental design, internal validity, variables tested, etc.

Validity of the findings

Please add p-values to the data Tables to indicate differences between heights, strategies, or interaction effects.

Additional comments

The authors have done a good job responding to reviewer comments. Please have a native English speaker give it a review and correct any language details that need to be changed.

·

Basic reporting

The manuscript is clear and the authors did a commendable job on replying to my comments with thoughful clarity and precision. The manuscript, in my opinion, has improved significantly and is enhanced by the additional data (on work and kinematics). The wording is also more carefully chosen and important references on statements relating to muscle-tendon dynamics have been added, removing the "speculation" from many of these statements.

1 concern remains: I disagree with the authors' statement that the ankle has less feed-forward control. There is plenty evidence that the lower leg muscles are pre-activated (E.g., Dick et al., 2021l Lichtwark et al., 2005; Aeles et al., 2017). I think this needs to be considered in the statements made in the current manuscript on this (only reference to opossite is in birds, which have slightly different (but also very similar) gait mechanics than humans (also at muscle-tendon level).

Experimental design

/

Validity of the findings

/

Additional comments

/

---

## Round 0.3 · accepted · Accept

Thank you for addressing the remaining reviewer concerns, and for improving the English grammar.\\